# Identification and Characterization of a Double-Stranded RNA Degrading Nuclease Influencing RNAi Efficiency in the Rice Leaf Folder *Cnaphalocrocis medinalis*

**DOI:** 10.3390/ijms23073961

**Published:** 2022-04-02

**Authors:** Jiajing Li, Juan Du, Shangwei Li, Xin Wang

**Affiliations:** Guizhou Provincial Key Laboratory for Agricultural Pest Management of Mountainous Regions, Institute of Entomology, Guizhou University, Guiyang 550025, China; jjli0315@163.com (J.L.); juandudj@163.com (J.D.); w727159525@163.com (X.W.)

**Keywords:** *Cnaphalocrocis medinalis*, RNA interference, RNAi efficiency, dsRNA degrading nuclease, dsRNase

## Abstract

Rice leaf folder *Cnaphalocrocis medinalis* is one of the most serious pests of rice in rice-planting regions worldwide. DsRNA-degrading nucleases (dsRNases) are important factors in reducing the efficiency of RNA interference (RNAi) in different insects. In this study, a *dsRNase* gene from *C. medinalis* (*CmdsRNase*) was cloned and characterized. The *CmdsRNase* cDNA was 1395 bp in length, encoding 464 amino acids. The CmdsRNase zymoprotein contains a signal peptide and an endonuclease NS domain that comprises six active sites, three substrate-binding sites, and one Mg^2+^-binding site. The mature CmdsRNase forms a homodimer with a total of 16 α-helices and 20 β-pleated sheets. Homology and phylogenetic analyses revealed that CmdsRNase is closely related to dsRNase2 in *Ostrinia nubilalis*. Expression pattern analysis by droplet digital PCR indicated that the expression levels of *CmdsRNase* varied throughout the developmental stages of *C. medinalis* and in different adult tissues, with the highest expression levels in the fourth-instar larvae and the hemolymph. CmdsRNase can degrade dsRNA to reduce the efficiency of RNAi in *C. medinalis*. Co-silencing of *CmCHS* (*chitin synthase* from *C. medinalis*) and CmdsRNase affected significantly the growth and development of *C. medinalis* and thus improved RNAi efficacy, which increased by 27.17%. These findings will be helpful for green control of *C. medinalis* and other lepidopteran pests by RNAi.

## 1. Introduction

RNA interference (RNAi) is a highly conserved mechanism triggered by double-stranded RNA (dsRNA) in the evolutionary process; therefore, homologous mRNAs are degraded efficiently and specifically. The essence of RNAi is post-transcriptional gene silencing. The transcription of the silenced genes continues to proceed normally, but the transcribed messenger RNA (mRNA) undergoes sequence-specific degradation in the cytoplasm, with the result that these genes cannot be normally expressed as proteins [1]. RNAi exists in most eukaryotes, but the efficiency of RNAi varies greatly among different species [2,3,4,5]. RNAi has high efficiency, specificity, and transmissibility, and is widely used as a powerful tool in the exploration of gene function analysis, biomedical research, biological pest control, and other fields. The use of RNAi technology to control pests is currently one of the hotspots in scientific research. The difference in RNAi efficiency among different species of insects limits the use of RNAi technology in basic insect research and pest control; for example, the RNAi efficiency in most coleopteran insects is high and long-lasting [6,7,8,9], whereas the RNAi efficiency in most dipteran, hemipteran, and lepidopteran insects is variable and unstable [10,11,12]. There are many factors that affect the efficiency of RNAi in insects, including delivery methods [13,14,15], dsRNA transport in cells [16], target genes [4,6], and tissues [17,18]. At present, there is evidence that maintaining the integrity of dsRNA before entering cells is a key factor in ensuring the efficiency of RNAi [19,20]. As dsRNA-degrading nuclease (dsRNase) can degrade dsRNA, researchers have focused on the molecular function of dsRNase when studying what affects the efficiency of RNAi.

Double-stranded ribonuclease, also known as dsRNase, belongs to the DNA/RNA non-specific endonuclease (NUC) family. NUCs are found in bacteria, viruses, nematodes, crustaceans, and mammals. In insects, NUCs include endonuclease G (EndoG) and dsRNases [21,22]. EndoG belongs to the single gene family in mitochondria and participates in mitochondrial DNA replication and repair [23,24], and in the case of cell apoptosis it is transferred into the nucleus to participate in DNA degradation. Additionally, dsRNase can degrade dsRNAs. Arimatsu et al. first studied the characteristic of dsRNase from the digestive juice of *Bombyx mori* in 2007; this type of dsRNase can degrade extracellular dsRNA, single-stranded RNA (ssRNA), and DNA [22,25]. Although dsRNase sequences have been cloned from many insects, their functional characteristics have been identified in only a few insects. At the present time, in addition to *B**. mori*, dsRNases from *Schistocerca gregaria*, *Locusta migratoria*, *Leptinotarsa decemlineata*, *Anthonomus grandis*, *Bemisia tabaci*, *Cylas puncticollis*, and *Spodoptera litura* have been identified as having the ability to degrade dsRNA [26,27,28]. Wynant et al. [26] identified four dsRNases from desert locust, specifically expressed in the midgut. After silencing *dsRNase2* with RNAi, the ability of the intestinal juice to degrade dsRNA weakened. Peng et al. [28] identified five dsRNases from *S. litura*, all of which have endonuclease_NS domains at the C-terminus; dsRNases1–4 contain a signal peptide at the N-terminus and dsRNase5 contains no signal peptide at the N-terminus. The activity of the zymoproteins expressed in the baculovirus expression system was determined and the results showed that the four dsRNases containing signal peptides have the ability to degrade dsRNA. These results showed that multiple dsRNases in *S. litura* caused the low efficiency and instability of RNAi in this insect.

Different insects respond differently to RNAi. Some insects have dsRNases in their intestines and hemolymph which degrade exogenous dsRNA that enters their own cells, thereby reducing the efficiency of RNAi [9,29,30]. Wang et al. [9] compared the differences in the efficacy of RNAi in four insects from various orders. The order of the sensitivity of insects to RNAi was as follows: *Periplaneta americana* > *Zophobas atratus* >> *L. migratoria* >> *S. litura*. *P**. americana* and *Z**. atratus* were sensitive to RNAi by injection and feeding, *L. migratoria* was sensitive to RNAi by injection, and *S. litura* was not sensitive to RNAi by injection and feeding. The hemolymph of *S. litura* degrades dsRNA at the fastest rate and the digestive juices of *S. litura* and *L. migratoria* can also quickly degrade dsRNA. The results indicated that the differences in the RNAi efficacy among these four insects were caused by the degradation of dsRNA in their bodies. Studies have shown that the efficiency of RNAi in some lepidopteran pests is low, mainly because of the presence of dsRNase in vivo, which degrades the incoming dsRNA, thus severely restricting the application in pest control based on plant-mediated RNAi. However, some studies have revealed that when RNAi is used to silence target genes, RNAi efficiency can be significantly improved if dsRNase is silenced simultaneously [26]. When *LmCht10* or *LmCHS1* dsRNA was orally delivered to *L. migratoria*, simultaneously injecting ds*LmdsRNase2* increased the effect of RNAi [31]. For *L. decemlineata*, removal of dsRNase activity in the gut juice enhanced RNAi efficiency, but using the same method in *S. gregaria* did not improve the efficacy [32]. For *C. puncticollis*, the efficiency of RNAi by injection was higher than that of RNAi by feeding. Researchers such as Prentice [33] first injected *C. puncticollis* larvae with ds*CpdsRNase3* and then fed them with ds*Snf7* 4 days later; the results demonstrated that larval mortality increased significantly and RNAi efficiency improved. Lepidopteran insects contain dsRNases that degrade exogenous dsRNA, resulting in reduced efficiency of RNAi by injection or feeding. If *dsRNase* genes are silenced while silencing target genes, RNAi efficiency will be significantly improved.

Rice leaf folder *Cnaphalocrocis medinalis* (Lepidoptera: Pyralidae) has been extensively reported as a notorious insect pest of rice in Asian rice-growing areas [34,35]. This species is a holometabolous insect, undergoing four developmental stages in its life cycle: egg, larva, pupa, and adult. *C. medinalis* larvae generally go through five instars and adults can migrate over long distances. This insect is a migratory rice pest and widely distributed in rice-planting regions worldwide [36]. Larvae feed on mesophyll tissues of rice leaves, which severely affects the photosynthesis of rice, leading to approximately 10–20% yield loss or even total crop failure. In this study, *dsRNase* from *C. medinalis*, named *CmdsRNase*, was cloned with reverse transcription-polymerase chain reaction (RT-PCR) and characterized with bioinformatics methods and digital PCR (dPCR). Functional analysis of *CmdsRNase* was performed using RNAi technology.

## 2. Results

### 2.1. Characteristics of CmdsRNase

The open reading frame (ORF) of the *CmdsRNase* cDNA is 1395 bp in length, encoding 464 amino acids (Figure 1). The molecular formula of CmdsRNase was C_2330_H_3534_N_676_O_663_S_17_, with a molecular weight of 52.17 kDa and a theoretical isoelectric point (pI) of 8.88. This zymoprotein contains a signal peptide of 16 amino acids at the N-terminus. Functional domain analysis revealed that CmdsRNase possesses an endonuclease NS domain (Figure 2a) that comprises six active sites, three substrate-binding sites, and one Mg^2+^-binding site (Figure 2b). The mature CmdsRNase contains four O-glycosylation (S8, S123, S140, and S208) and two N-glycosylation sites (N36 and N297). CmdsRNase includes 36 negatively charged amino acid residues (Asp + Glu) and 43 positively charged amino acid residues (Arg + Lys), with a calculated instability index (II) of 39.05, which indicates that it is a stable protein since a protein with an II greater than 40 is unstable. The aliphatic index of CmdsRNase was predicted to be 75.71 and the grand average of hydropathicity (GRAVY) to be −0.31. Homology modeling showed that the mature CmdsRNase (amino acids 159–440) forms a homodimer, with a total of 16 α-helices, 20 β-pleated sheets, and 36 random coils (Figure 3a). As shown in Figure 3b, deep red indicates the evolutionarily conserved amino acids that are of vital importance in the structure and function of CmdsRNase. 

### 2.2. Homology Comparison and Cluster Dendrogram

CmdsRNase showed 75.37% similarity with dsRNase2 from *O. nubilalis* based on BLAST against the NCBI NR database containing its amino acid sequence. The phylogenetic tree of 36 dsRNases from 25 insect species showed that dsRNases from the same order grouped into a clade and CmdsRNase clustered together with dsRNA-degrading nuclease from *O. nubilalis* (Figure 4). These results indicate that dsRNase is conserved during evolution and that *C. medinalis* is the cloest relative of *O. nubilalis*.

### 2.3. Gene Expression Profiles

The droplet digital PCR (ddPCR) results showed that *CmdsRNase* was expressed in the larvae, with the highest level in the fourth-instar larvae, and was almost not expressed in the eggs, pupae, and adults (Figure 5). This gene was expressed in the tissues tested from *C. medinalis* adults, with the highest level in the hemolymph, followed by the midgut, and with the lowest level in the head and integument. The *CmdsRNase* expression level in the hemolymph was 59 times that in the head and integument (Figure 6).

### 2.4. Degradation of dsRNA by Crude CmdsRNase 

The agarose gel electrophoresis assay showed that 120 ng of dsRNA was totally degraded by 10 μg of crude CmdsRNase in 1 min at 28 °C (Figure 7a) and by 5 μg of crude CmdsRNase in 5 min at 28 °C (Figure 7b). The results indicated that the crude CmdsRNase possessed the ability to degrade dsRNA.

### 2.5. Effect of dsCmdsRNase Injection on RNAi Efficiency

To further verify the function of dsRNase in *C. medinalis*, dsRNAs of *CmRNase* and *CmCHS* (*chitin synthase* gene from *C. medinalis*) were jointly injected into the third-instar larvae to verify whether this enzyme can degrade dsRNA in the hemolymph. The expression level of *CmCHS* after co-injection of ds*CmdsRNase* + ds*CmCHS* was significantly lower than that of *CmCHS* at the third and fourth days after injection of ds*CmCHS* alone. The *CmCHS* expression level at the third day post co-injection was reduced by 2 times compared with the ds*CmCHS*-injected group, in which the *CmCHS* level was reduced by 2.3 times compared with the control. The *CmCHS* level at the fourth day post co-injection decreased by 2.4 times compared with the ds*CmCHS*-injected group, in which the *CmCHS* level decreased by 2 times compared with the control (Figure 8). Three days after co-injection of ds*CmdsRNase* + ds*CmCHS* and injection of ds*CmCHS*, RNAi efficiency of *CmCHS* in *C. medinalis* reached 78.04% and 56.84%, respectively; this efficiency increased by 27.17% (Figure 9). The corrected mortality of the ds*CmCHS*-injected larvae (53.3%) was 6.4 times that of the ds*GFP*-injected larvae (8.3%) at the seventh day after injection, and the corrected mortality of larvae injected with ds*CmdsRNase* + ds*CmCHS* (81.7%) was 1.5 times that of the ds*CmCHS*-injected larvae (53.3%), whereas ds*GFP* had little influence on larval survival (Figure 10).

### 2.6. Effect of RNAi on Phenotypes of C. medinalis

After *CmCHS* was silenced, phenotypic changes of *C. medinalis* were recorded. Larvae injected with ds*CmCHS* or ds*CmdsRNase* + ds*CmCHS* displayed phenotypic alterations, such as lower vitality, smaller body size, and lighter weight. The abdomen of larvae shrank (Figure 11D), the body color turned black (Figure 11E), and the head got bigger (Figure 11D,F). Some larvae with *CmCHS* gene knockdown died. In addition, *CmCHS* RNAi larvae could not molt and pupate normally; however, the larvae in the control group grew normally and there were no obvious phenotypic changes. Some pupae from larvae with *CmCHS* RNAi blackened and exhibited deformed phenotypes (Figure 12). RNAi also caused adult deformities, such as abnormal wing folding and unfolding (Figure 13B,C), and malformation, with pupa-shaped abdomens observed in adults (Figure 13D). In the experimental group injected with ds*CmdsRNase* + ds*CmCHS*, it was observed that there were normal-sized adults without wing deformities, but they had no ability to fly. These results indicate that co-silencing of both *CmCHS* and *CmdsRNase* can lead to serious developmental disorders and death of *C. medinalis*.

## 3. Discussion

The analysis of the overall expression pattern of *CmdsRNase*, especially the high expression in the hemolymph and midgut of the larvae, showed that the activity of the nuclease increased during the larval stage, and the high level of dsRNase in the hemolymph and midgut may help reduce larval RNAi efficiency. Recent research on *S**. litura* supports this view [28]. However, in *Spodoptera exigua*, the nuclease activity of the intestinal supernatant at different developmental stages is relatively low, and the lower level of dsRNase may be one of the factors leading to the higher RNAi response of this insect [37]. With tissue specificity, almost all insect dsRNases are highly expressed in the intestine and hemolymph [31,33]; the expression profile of *CmdsRNase* in *C. medinalis* is consistent with the general pattern, indicating the function of CmdsRNase to degrade dsRNA that enters the gut lumen or hemolymph.

It has been reported that LmdsRNase1 in *L**. migratoria* can effectively degrade dsRNA at pH5 and is highly expressed in blood cells, but the physiological pH value of hemolymph (7.0) strongly inhibits the activity of *LmdsRNase1*, making dsRNA stable in the hemolymph for a long time. Therefore, although dsRNase is expressed in different tissues, some may not degrade dsRNA due to physiological pH or substrate-specific factors [26]. Our results showed that the hemolymph and the crude CmdsRNase extracted from the hemolymph had similar ability to degrade dsRNA; this may be because the hemolymph contained dsRNases that degrade dsRNA. Although *CmdsRNase* is mainly expressed in the hemolymph, dsRNase in the hemolymph of *C. medinalis* is active, which is consistent with the results of the study on dsRNase in the hemolymph of *Manduca sexta* [38]. The conditions for dsRNase to degrade dsRNA are different in different insects.

In insects, dsRNases reduce RNAi efficiency by degrading dsRNA. In *S**. gregaria*, interfering with *dsRNase2* improved RNAi efficiency, while interfering with other *dsRNases* had no effect [26,32]. In *C**. puncticollis*, only dsRNase3 could affect RNAi efficiency [33]; however, in *S**. litura*, the combined action of multiple dsRNases led to a decrease in RNAi efficiency [28,30], indicating that the number of dsRNases and their mechanisms of action may be different in various insects. In the present study, by co-injecting ds*CmCHS* and ds*CmRNase* + ds*CmCHS* in *C. medinalis*, the degrading activity of *CmdsRNase* on dsRNA was confirmed and the RNAi efficiency in this pest was significantly elevated. This experiment was only a preliminary study on the function of a single dsRNase; it does not rule out the existence of a synergistic effect between this enzyme and other dsRNases of *C. medinalis*. In the next step, we will try to silence multiple enzymes simultaneously so as to further improve the efficacy of RNAi in *C. medinalis*.

## 4. Materials and Methods

### 4.1. Insect-Rearing and Sample Preparation

*C. medinalis* was collected from a rice field in Guiyang, Guizhou, China, and maintained in an insectary of the Institute of Entomology, Guizhou University, at 26 ± 1 °C and 75 ± 5% relative humidity under a 14:10 h light: dark photoperiod. After three consecutive generations, insects at the same developmental stage were collected and stored in RNAlater (Qiagen, Duesseldorf, Germany) at −20 °C until use. These samples included eggs, first–fifth-instar larvae, pupa, adults, and imaginal tissues (the head, hemolymph, fat bodies, testis, ovary, midgut, and integument).

### 4.2. RNA Extraction and cDNA Synthesis

Total RNA was extracted at different developmental stages and from various imaginal tissues of *C. medinalis* using an Eastep Super Total RNA Kit (Promega, Madison, WI, USA) according to the manufacturer’s instructions. The quality and purity of the isolated RNA were determined by using agarose gel electrophoresis and a NanoDrop 2000 spectrophotometer (Thermo Fisher Scientific, Waltham, MA, USA), respectively. Then, cDNA was synthesized using a PrimeScript First-Strand cDNA Synthesis Kit (Takara Bio, Beijing, China) according to the manufacturer’s instructions and stored at −20 °C.

### 4.3. Cloning CmdsRNase 

Specific primers were designed based on the *CmdsRNase* sequence using Primer Premier 6.0 (PREMIER Biosoft, Palo Alto, CA, USA). The *CmdsRNase* ORF was then amplified using RT-PCR with cDNA from *C. medinalis* as a template. RT-PCR was performed with 2× Taq PCR Star Mix (GenStar, Beijing, China) under the following conditions: 2 min at 94 °C; 32 cycles of 30 s at 94 °C, 30 s at 55 °C, and 60 s at 72 °C; and 10 min at 72 °C. Subsequently, the PCR products were run on a 1% agarose gel for 25 min at 130 V, and the expected band was excised from the gel and purified using a DiaSpin DNA Gel Extraction Kit (Sangon Biotech, Shanghai, China). The recovered PCR products were cloned into a pMD18-T vector (Takara Bio, Dalian, China) and the recombinant pMD-18-T-CmdsRNase vector was transformed into *E. coli* DH5α competent cells (Invitrogen, Carlsbad, CA, USA), which were plated on Luria–Bertani agar plates and incubated at 37 °C overnight. Finally, the colony PCR was performed to identify positive clones that were submitted to Sangon Biotech (Shanghai, China) for sequencing.

### 4.4. Bioinformatic Analyses of CmdsRNase

The ORF of *CmdsRNase* was located with ORFfinder (https://www.ncbi.nlm.nih.gov/orffinder) (accessed on 6 May 2021). The molecular weight and isoelectric point of CmdsRNase were predicted using the Expasy ProtParam platform (https://web.expasy.org/protparam) (accessed on 6 May 2021). The signal peptide was predicted by SignalP-5.0 Server (https://services.healthtech.dtu.dk/service.php?SignalP-5.0) (accessed on 6 May 2021). Domains of the zymoprotein were analyzed using SMART (http://smart.embl-heidelberg.de) and CDD (https://www.ncbi.nlm.nih.gov/cdd) (accessed on 6 May 2021). Prediction of glycosylation sites was carried out using the NetOGlyc 4.0 Server (https://services.healthtech.dtu.dk/service.php?NetOGlyc-4.0) and the NetNGlyc 1.0 Server (https://services.healthtech.dtu.dk/service.php?NetNGlyc-1.0) (accessed on 6 May 2021). The phylogenetic tree was constructed using the neighbor-joining method in MEGA X software with 1000 runs. The three-dimensional structure of the mature CmdsRNase was constructed with SWISS-MODEL homologous modeling (https://swissmodel.expasy.org) (accessed on 8 August 2021) and its molecular graph was drawn using PyMOL 2.5 (Schrodinger, New York, NY, USA).

### 4.5. Gene Expression Analyses Using ddPCR

The mRNA expression levels of *CmdsRNase* were detected using a QX200 Droplet Digital PCR system (ddPCR) (Bio-Rad, Hercules, CA, USA) at different developmental stages and in various adult tissues. The disposable eight-channel DG8 cartridge was placed in the cartridge holder, and 20 μL of PCR mixtures were transferred to the middle wells of the cartridge. The lower wells were filled with 70 μL of droplet-generation oil. The cartridge containing the PCR mixtures and oil was placed into a Droplet Generator (Bio-Rad, Hercules, CA, USA) to generate individual droplets. Then, 40 μL of droplets were transferred into wells of a 96-well PCR plate, which was heat-sealed at 180 °C for 5 s with a permeable foil using a PX1 PCR Plate Sealer and loaded into a C1000 Touch Thermal Cycler (Bio-Rad, Hercules, CA, USA). The PCR reaction system and conditions are listed in Table 1. After PCR was complete, the sealed plate was placed into a Droplet Reader (Bio-Rad, Hercules, CA, USA) to count the positive and negative droplets. The ddPCR experiment was repeatedly performed three times for each sample. Data were analyzed using QuantaSoft software (Bio-Rad, Hercules, CA, USA) and SPSS 22.0 (SPSS Inc., Chicago, IL, USA).

### 4.6. Crude CmdsRNase Extraction and dsRNA Degrading Assay

*C. medinalis* larvae were torn using dissecting forceps and put into a 0.8 mL centrifuge tube with four holes at its bottom. This small tube was then put into a 1.5 mL tube containing 1 mM phenylthiourea and 1 mM phenylmethylsulfonyl fluoride (PMSF). The double-tube device was centrifuged at 2500× *g* for 10 min at 4 °C and the collected hemolymph was centrifuged at 12,000× *g* for 10 min at 4 °C to remove cells for obtaining serum. Next, 0.1 M NaOH was added to the serum to adjust the pH to 8.8 (the isoelectric point of CmdsRNase), and the precipitate was collected by centrifugation at 12,000× *g* for 10 min at 4 °C and dissolved with 0.01 M phosphate-buffered saline (PBS) to obtain crude CmdsRNase. 

The concentration of the crude CmdsRNase zymoprotein was measured using a BCA Protein Quantification Kit (Yesea, Shanghai, China) according to the manufacturer’s instructions in a Multiskan GO (Thermo Fisher Scientific, Waltham, MA, USA). For an in vitro incubation assay, 1 μL of ds*CmCHS* solution (containing 120 ng of dsRNA, 381 bp) was mixed with 10 μg of the crude CmdsRNase in 1.5 mL centrifuge tubes and incubated at 28 °C for 1, 10, 20, 4, and 60 min, respectively. Then, 120 ng of ds*CmCHS* was mixed separately with 1, 5, 10, 15, and 20 μg of the crude CmdsRNase in 1.5 mL centrifuge tubes and incubated at 28 °C for 5 min. After incubation, these samples were subjected to 1.5% agarose gel electrophoresis to evaluate the integrity of residual dsRNA.

### 4.7. RNA Interference

According to the ORFs of *CmdsRNase* and *CmCHS* (*chitin synthase* gene from *C. medinalis*), two online RNAi design tools, including siDirect (http://sidirect2.rnai.jp) and DSIR (http://biodev.extra.cea.fr/DSIR/DSIR.html) (accessed on 6 May 2021), were used to search for fragments targeting *CmdsRNase* and *CmCHS* mRNAs. The *green fluorescent protein* gene (*GFP*) (GenBank: CAA58789) from *Aequorea victoria* was used as the internal control. The target fragments were amplified using CmdsRNase-iF/CmdsRNase-iR, CmCHS-iF/CmCHS-iR, and GFP-iF/GFP-iR primers, respectively. After purification, the PCR products were inserted into the pMD18-T vector (Takara Bio, Dalian, China) for sequencing. Clones containing the correct sequences were cultured. Plasmids were extracted and used as templates to separately amplify target fragments with CmdsRNase-dsF/CmdsRNase-dsR, CmCHS-dsF/CmCHS-dsR, and GFP-dsF/GFP-dsR primers. After the PCR product was purified, DNA at a high concentration (not less than 300 ng/μL) was used for the template-synthesis of ds*CmdsRNase*, ds*CmCHS*, and ds*GFP* using a TranscriptAid T7 High Yield Transcription Kit (Thermo Fisher Scientific, Waltham, MA, USA). Briefly, the in vitro transcription system (20 μL) contained 2 μL of nuclease-free water, 4 μL of 5 × reaction buffer, 8 μL of ATP/CTP/GTP/UTP mix, 4 μL of DNA template (with T7 at both ends), and 2 μL of enzyme mix. After vortexing and briefly spinning, the tubes were incubated at 37 °C for 6 h. The dsRNA was purified using a GeneJET RNA Purification Kit (Thermo Fisher Scientific, Waltham, MA, USA) and then detected by agarose gel electrophoresis and a NanoDrop 2000 spectrophotometer (Thermo Fisher Scientific, Waltham, MA, USA) to evaluate its integrity and quality. The primers used in this study are listed in Table 2.

Twenty healthy third-instar larvae were selected for each RNAi experiment in each group. One-point-five micrograms of ds*CmCHS* or the mixture of ds*CmdsRNase* and ds*CmCHS* were injected into the eighth abdominal segment of each larva using a Nanoliter 2020 Injector (World Precision Instruments, Sarasota, FL, USA), then the larvae were moved onto fresh rice leaves inside glass tubes and cultured in an artificial climate box. The feeding, phenotype, and survival of the larvae were observed each day, and the leaves were replaced with fresh ones every two days. Larvae injected with the same amount of ds*GFP* were used as the control. These experiments were replicated 4 times for each group. Two surviving larvae were collected from each group every day over four days to detect the expression level of *CmCHS* by ddPCR.

### 4.8. Statistical Analyses

Data are expressed as the means ± SD from at least three independent experiments. Statistical analyses were performed with one-way analysis of variance followed by Duncan’s multiple range test using SPSS 22.0 (SPSS Inc., Chicago, IL, USA). Statistical significance was set at *p* < 0.05.

## 5. Conclusions

In this study, a *dsRNase* in *C. medinalis* was cloned and characterized using PCR and bioinformatics technologies. CmdsRNase can degrade dsRNA to reduce the efficiency of RNAi in *C. medinalis*. Co-silencing of the target genes *CmCHS* and *CmdsRNase* can significantly improve the RNAi effect on *C. medinalis.* This research provides a new strategy for RNAi-mediated insect pest control and will be helpful in promoting green control of lepidopteran pests by RNAi.

## Figures and Tables

**Figure 1 ijms-23-03961-f001:**
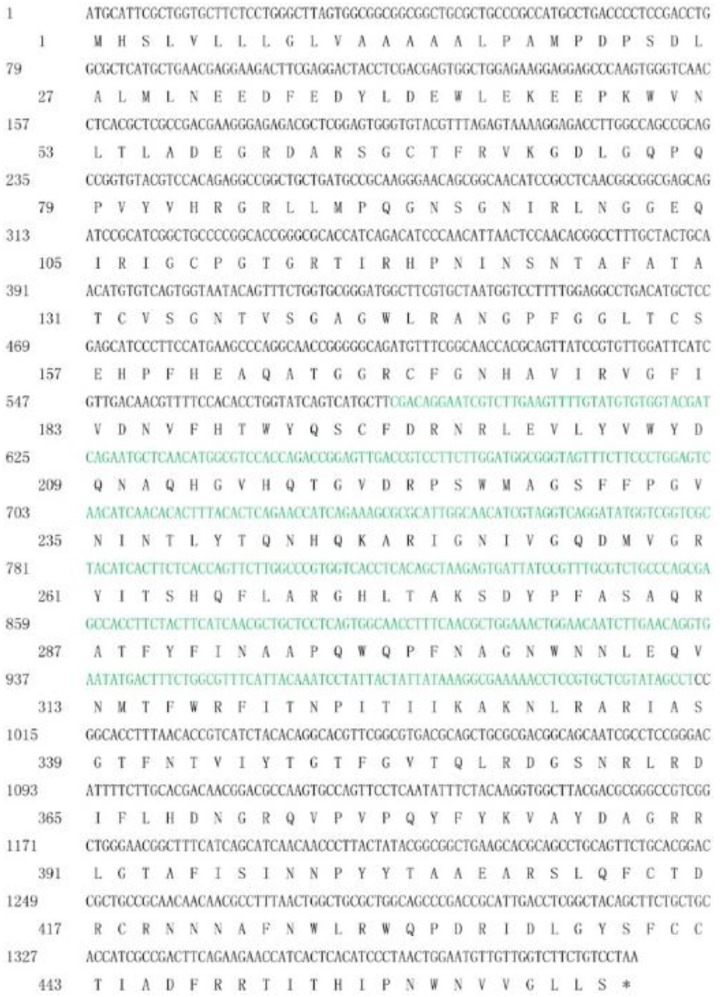
Nucleotide and deduced amino acid sequences of *CmdsRNase* (*dsRNase* from *C. medinalis*). The asterisk indicates the stop codon, and the target sequence of RNAi is marked in green (585–1012 bp).

**Figure 2 ijms-23-03961-f002:**
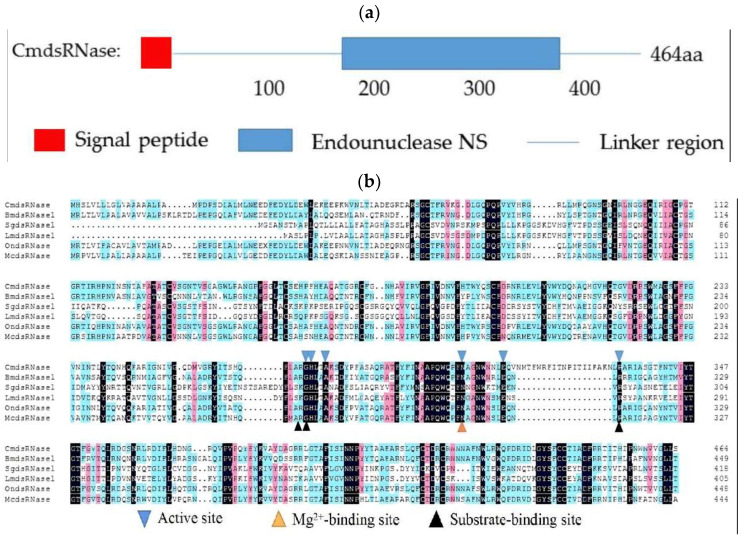
Analyses of the amino acid sequence of CmdsRNase (dsRNase from *C. medinalis*). (**a**) Schematic diagram of the domain of CmdsRNase. The red box, blue box, and blue line represent the signal peptide, endonuclease_NS domain, and linker region, respectively. (**b**) Multiple sequence alignment of dsRNases from different insect species: CmdsRNase (*C**. medinalis*), BmdsRNase (*B**. mori*, BAF33251.1), SgdsRNase (*S**. gregaria*, AHN55088.1), LmdsRNase (*L**. migratoria*, ARW74135.1), OndsRNase (*Ostrinia nubilalis*, MT524712.1), and McdsRNase (*Mamestra configurata*, HM357845.1). Blue inverted triangles represent active sites of six key amino acid residues, the yellow triangle indicates Mg*^2+^*-binding sites, and black triangles mark substrate-binding sites.

**Figure 3 ijms-23-03961-f003:**
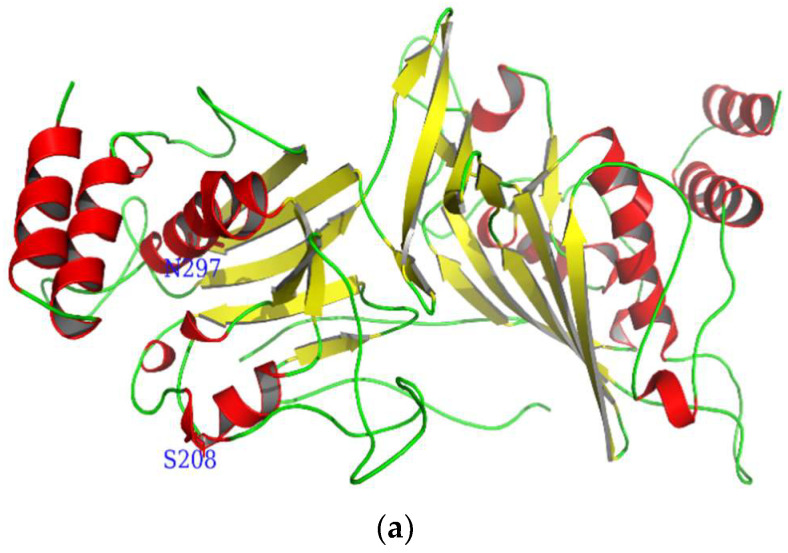
Three-dimensional molecular structure of CmdsRNase. (**a**) This graphic was drawn using PyMOL 2.5 based on the CmdsRNase.pdb data. Red represents α-helices, yellow indicates β-pleated sheets, and green denotes random coils. S208 and N297 indicate O- and N-glycosylation, respectively. (**b**) The highly conserved region (in deep red) is displayed in the structure. The homology model was performed by the ConSurf software (https://consurf.tau.ac.il) (accessed on 20 December 2021) and optimized using PyMOL 2.5.

**Figure 4 ijms-23-03961-f004:**
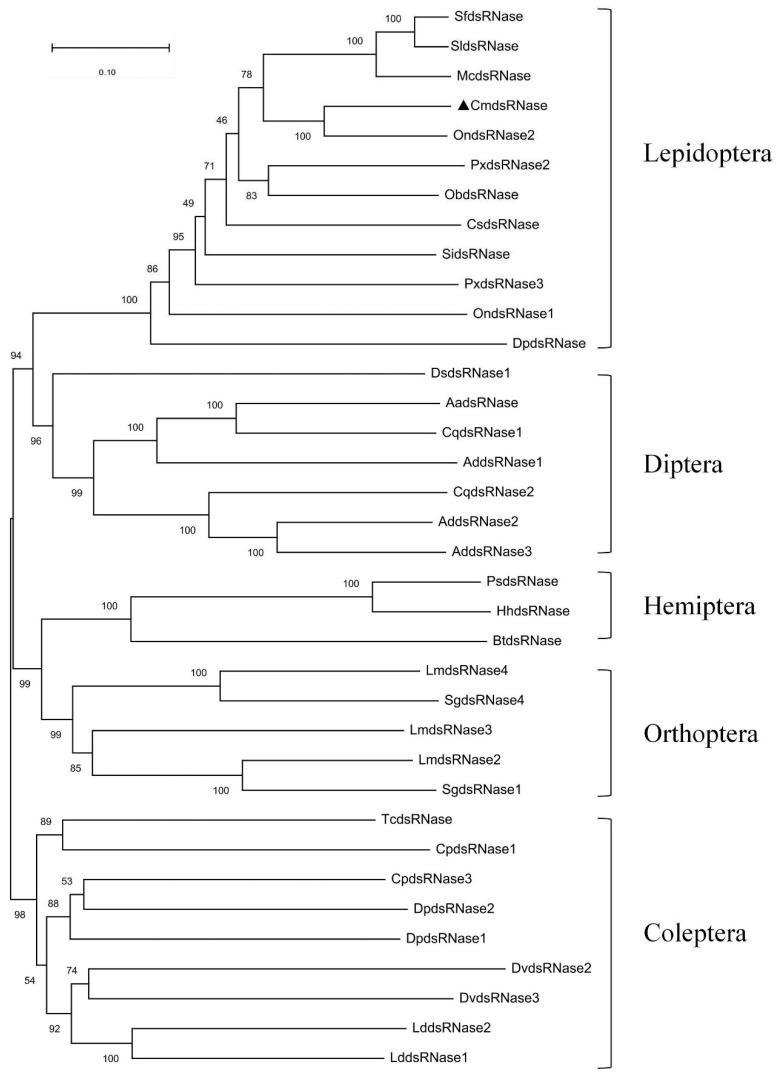
An evolutionary tree of insect dsRNases. Bootstrap values are marked at internal nodes and CmdsRNase is labeled with a solid triangle. The species name and GenBank accession number corresponding to each sequence are as follows. Lepidoptera: SfdsRNase, *Spodoptera frugiperda* (CAR92521.1); SldsRNase, *S**. litura* (QJD55608.1); McdsRNase, *M**. configurata* (AEA76311.1); OndsRNase1, *O**. nubilalis* (QOE54913.1); OndsRNase2, *O**. nubilalis* (QOE54910.1); PxdsRNase2, *Plutella xylostella* (QZW25238.1); PxdsRNase3, *P**. xylostella* (QZW25239.1); ObdsRNase, *Operophtera brumata* (KOB65521.1); CsdsRNase, *Chilo suppressalis* (AKB95584.1); SidsRNase, *Sesamia inferens* (AKB95590.1); DpdsRNase, *Danaus plexippus*
*plexippus* (OWR44806.1). Diptera: DsdsRNase1, *Drosophila suzukii* (QXY82428.1); AadsRNase, *Aedes aegypti* (EAT42072.1); CqdsRNase1, *Culex quinquefasciatus* (EDS34867.1); CqdsRNase2, *C**. quinquefasciatus* (EDS38458.1); AddsRNase1, *Anopheles darlingi* (ETN62076.1); AddsRNase2, *A**. darlingi* (ETN61460.1); AddsRNase3, *A**. darlingi* (ETN61459.1). Orthoptera: LmdsRNase2, *L**. migratoria* (ARW74135.1); LmdsRNase3, *L**. migratoria* (ARW74136.1); LmdsRNase4, *L**. migratoria* (ARW74137.1); SgdsRNase1, *S**. gregaria* (AHN55088.1); SgdsRNase4, *S**. gregaria* (AHN55091.1); Coleptera: CpdsRNase1, *C**. puncticollis* (QCF41178.1); CpdsRNase3, *C**. puncticollis* (QCF41177.1); DvdsRNase2, *Diabrotica virgifera virgifera* (QNH88358.1); DvdsRNase3, *D**. virgifera* (QNH88359.1); TcdsRNase, *Tribolium castaneum* (QJD55726.1); LddsRNase1, *L**. decemlineata* (APF31792.1); LddsRNase2, *L**. decemlineata* (APF31793.1); DpdsRNase1, *Dendroctonus ponderosae* (ENN82866.1); DpdsRNase1, *D**. ponderosae* (ERL84039.1); Hemiptera: PsdsRNase, *Plautia stali* (BCL51433.1); BtdsRNase, *Bemisia tabaci* (AQU43107.1); HhdsRNase, *Halyomorpha halys* (XP_014282547.1).

**Figure 5 ijms-23-03961-f005:**
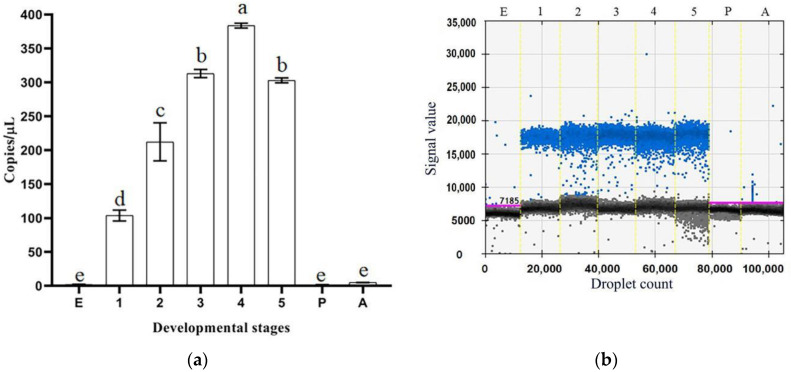
Expression levels of *CmdsRNase* at different developmental stages of *C. medinalis*. (**a**) Histogram. (**b**) Droplet generation diagram. Blue and black dots indicate positive and negative, respectively. E, Egg; 1–5, first–fifth-instar larvae; P, Pupa; A, Adult. Each bar represents the mean ± SD. Different letters above the bars indicate significant differences at *p* < 0.05 based on Duncan’s test.

**Figure 6 ijms-23-03961-f006:**
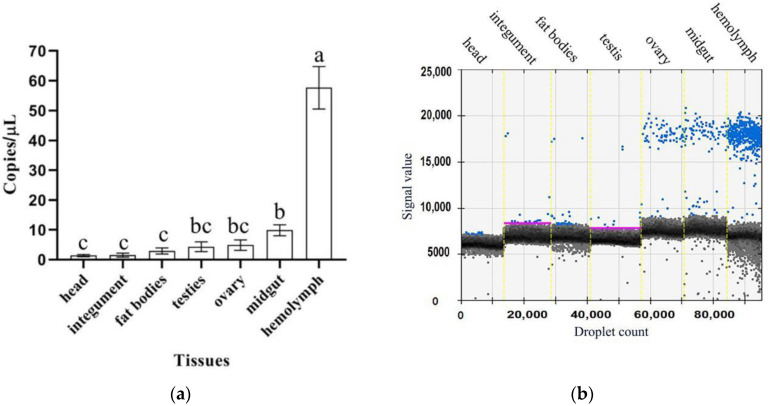
Expression levels of *CmdsRNase* in various tissues of *C. medinalis* adults. (**a**) Histogram. (**b**) Droplet generation diagram. Blue and black dots indicate positive and negative, respectively. Each bar represents the mean ± SD. Different letters above the bars indicate significant differences at *p* < 0.05 based on Duncan’s test.

**Figure 7 ijms-23-03961-f007:**
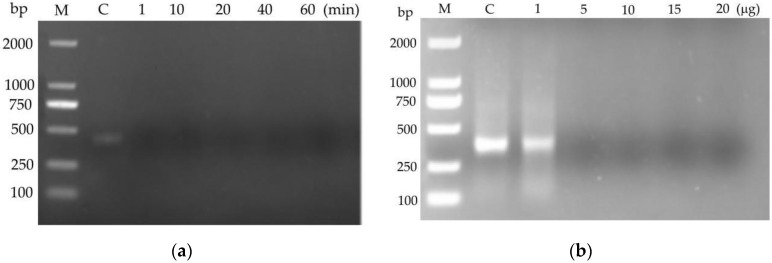
Degradation of dsRNA by the crude CmdsRNase. (**a**) Degradation of dsRNA (120 ng) by 10 μg crude CmdsRNase at different times. (**b**) Degradability of dsRNA (120 ng) by different quantities of crude CmdsRNase within 5 min. M: DL2000 DNA marker; C: Control.

**Figure 8 ijms-23-03961-f008:**
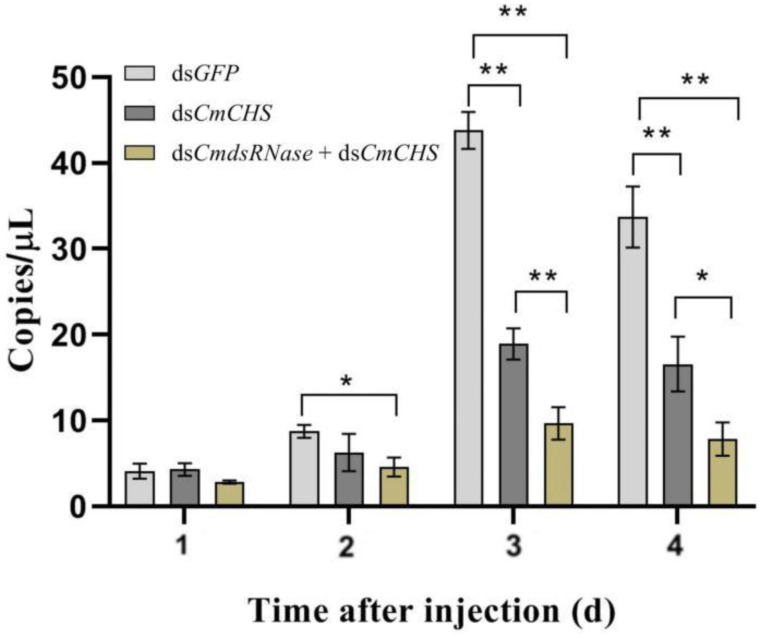
Expression levels of *CmCHS* at different times after dsRNA injection. The *CmCHS* expression levels in the third-instar *C. medinalis* larvae were detected with ddPCR four days after injection of ds*CmCHS* or the mixture of ds*CmdsRNase* and ds*CmCHS*. Larvae injected with ds*GFP* were used as the control group. Each bar represents the mean ± SD. One asterisk above bars indicates a significant difference at *p* < 0.05 and two asterisks indicate a very significant difference at *p* < 0.01 according to Duncan’s test.

**Figure 9 ijms-23-03961-f009:**
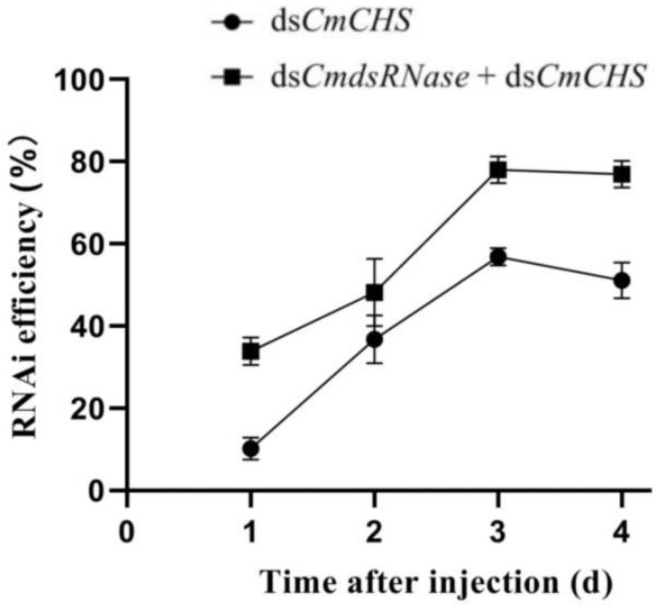
RNAi efficiency at different times. RNAi efficiency in the third-instar *C. medinalis* larvae was calculated four days after injection of ds*CmCHS* or the mixture of ds*CmdsRNase* and ds*CmCHS.* Data are shown as the means ± SD.

**Figure 10 ijms-23-03961-f010:**
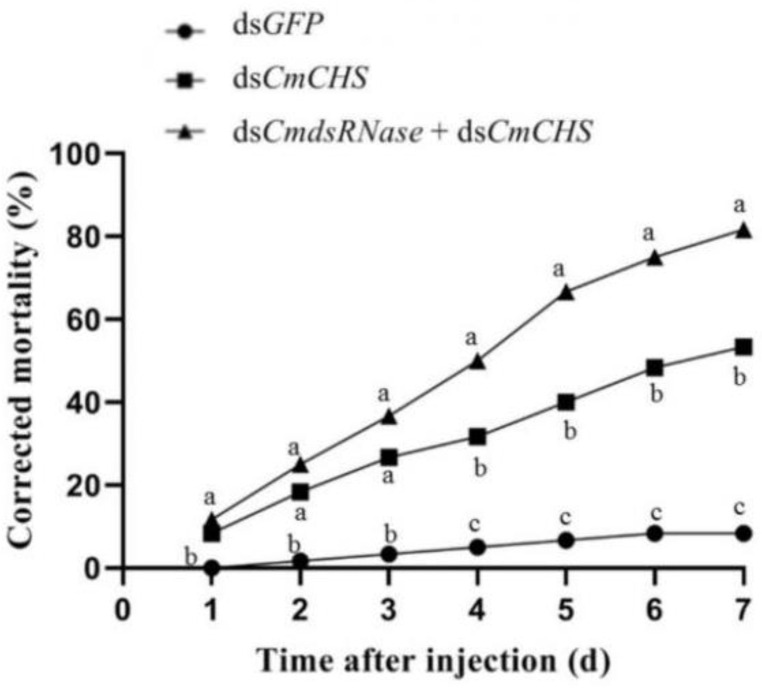
Corrected mortality of larvae within seven days after dsRNA injection. Corrected mortality of the third-instar *C. medinalis* larvae was calculated within seven days after injection with ds*CmCHS* or the mixture of ds*CmdsRNase* and ds*CmCHS.* Larvae injected with ds*GFP* were used as the control group. Different letters above the broken lines indicate significant differences between treatments on the same day (*p* < 0.05, Duncan’s test).

**Figure 11 ijms-23-03961-f011:**
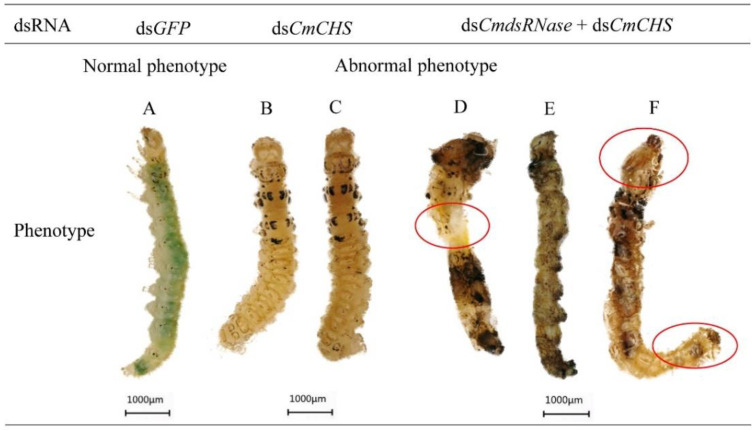
Abnormal morphology of *C. medinalis* larvae after dsRNA injection. (**A**) Phenotypes of larvae after ds*GFP* injection. (**B**,**C**) ds*CmCHS*-injected larvae. (**D**–**F**) Larvae injected with ds*CmdsRNase* + ds*CmCHS*. Red circles indicate the phenotypic changes in the larvae. Each scale bar represents 1000 μm.

**Figure 12 ijms-23-03961-f012:**
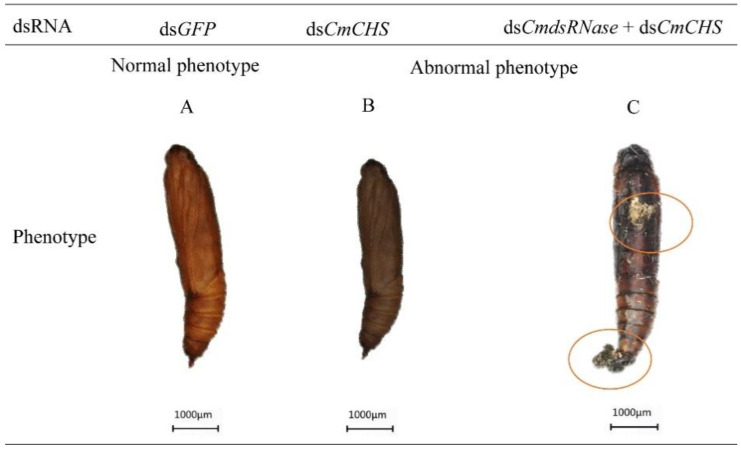
Abnormal morphology of *C. medinalis* pupae after dsRNA injection. (**A**) ds*GFP*-injected pupa. (**B**) Darkened pupa. (**C**) Malformed pupa. Red circles indicate deformed changes in the pupa. Each scale bar represents 1000 μm.

**Figure 13 ijms-23-03961-f013:**
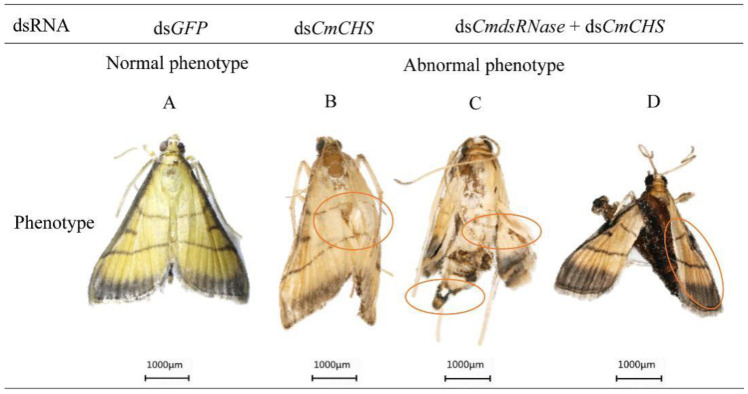
Abnormal morphology of *C. medinalis* adults after dsRNA injection. (**A**) ds*GFP*-injected adult. (**B**) Abnormal wing unfolding. (**C**) Wing and terminal abdominal deformities. (**D**) Adult deformity with pupa-shaped abdomen. Malformed phenotypes are marked with red circles. Each scale bar represents 1000 μm.

**Table 1 ijms-23-03961-t001:** Reaction system and procedure of droplet digital PCR.

Component	Volume per Reaction, μL	Final Concentration	Cycling Step	Temperature, °C	Time	Ramp Rate	Cycles
2× QX200 ddPCR EvaGreen Supermix	10	1×	Enzyme activation	95	5 min	2 °C/s	1
Forward primer (2 μM)	1	100 nM	Denaturation	95	30 s		40
Reverse primer (2 μM)	1	100 nM	Annealing and extension	60	1 min		
Diluted cDNA template	1	200 ng/μL	Signal stabilization	4	5 min		1
DNase-free water	7	-		90	5 min		1
Total volume	20	-	Hold	4	Infinite		1

Note: Use a heated lid set to 105 °C and set the sample volume to 40 µL.

**Table 2 ijms-23-03961-t002:** Primers used for cloning and expression analysis of *Cm**dsRNase* from *C. medinalis*.

Primer Name	Primer Sequence (5′→3′)	Primer Usage
CmdsRNase-F	ATGCATTCGCTGGTGCTTC	RT-PCR
CmdsRNase-R	TTAGGACAGAAGACCAACAAC	
CmdsRNase-dF	GACGCCAAGTGCCAGTTCCT	ddPCR
CmdsRNase-dR	GTGCTTCAGCCGCCGTATAGT	
CmCHS-dF	TGGAATACCTTCGCCAGTCATC	
CmCHS-dR	CCAGGAACACCAGGAGGCATT	
CmdsRNase-iF	CGACAGGAATCGTCTTGAAG	dsRNA synthesis
CmdsRNase-iR	AGGCTATACGAGCACGGAGGT	
CmdsRNase-dsF	taatacgactcactatagggCGACAGGAATCGTCTTGAAG	
CmdsRNase-dsR	taatacgactcactatagggAGGCTATACGAGCACGGAGGT	
CmCHS-iF	ACGAGGTTACACGAGAGG	
CmCHS-iR	CATCCAATGTTCCAATGTTCCT	
CmCHS-dsF	taatacgactcactatagggACGAGGTTACACGAGAGG	
CmCHS-dsR	taatacgactcactatagggCATCCAATGTTCCAATGTTCCT	
GFP-iF	GCCAACACTTGTCACTACTT	
GFP-iR	GGAGTATTTTGTTGATAATGGTCTG	
GFP-dsF	taatacgactcactatagggGCCAACACTTGTCACTACTT	
GFP-dsR	taatacgactcactatagggGGAGTATTTTGTTGATAATGGTCTG	

Note: The lowercase letters in the primer sequences represent the sequence of the T7 promoter.

## Data Availability

Not applicable.

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
