# Peer review of "Identification and Characterization of a Double-Stranded RNA Degrading Nuclease Influencing RNAi Efficiency in the Rice Leaf Folder Cnaphalocrocis medinalis"

_ijms, 2022, doi:10.3390/ijms23073961_

Round 1

Reviewer 1 Report

The MS of Jia-Jing Li et al presents some interesting data about RNAi efficiently in combination of chitin synthetase and RNase silencing on rice leaf folder. 

There are some comments and suggestions:
2. Experimental design is not clear. I did not find information about effect/mortality under dsCmdsRNase in 2.5 section of results. 
3. Figure 10 and description in the results sections can contain stat analysis of synergetic action between dsCmCHS and dsCmdsRNase. But there is no information about mortality under dsCmdsRNase injection. 
4. The CmdsRNase needs to be decoded in figure legends (Fig 1-2).
5. Figure legends (8-10) should be improved to give readers better explanation of results (treatments, insect name etc.)
6. Conclusion should be added.
7. line 379 E=what is the concentration of PTU and PMSF?

Author Response

The MS of Jia-Jing Li et al presents some interesting data about RNAi efficiently in combination of chitin synthetase and RNase silencing on rice leaf folder. There are some comments and suggestions: 2. Experimental design is not clear. I did not find information about effect/mortality under dsCmdsRNase in 2.5 section of results. Response: We appreciate your valuable comment. We revised the content of the experimental design section and please kindly see Section 4 “Materials and Methods” in the revised manuscript. The information about effect/mortality after injection of dsCmdsRNase was added to Section 2.5 “Effect of dsCmdsRNase injection on RNAi efficiency”. 3. Figure 10 and description in the results sections can contain stat analysis of synergetic action between dsCmCHS and dsCmdsRNase. But there is no information about mortality under dsCmdsRNase injection. Response: We modified Figure 10 and its caption. Please kindly see Lines1186-1190 in the revised manuscript. The corrected mortality rate of co-injection of dsCmdsRNase+dsCmCHS was significantly higher than that of single injection of dsCmCHS, indicating that co-injection can greatly improve the RNAi efficiency in C. medinalis. The mortality rate of dsCmdsRNase injection alone was not significant. The study is focused on how CmdsRNasw affects the RNAi efficiency in C. medinalis. 4. The CmdsRNase needs to be decoded in figure legends (Fig 1-2). Response: We added the explanation of CmdsRNase in titles of Figures 1 and 2. Please see Lines 741 and 751 in the revised manuscript. CmdsRNase in Figure represents the gene (in italics) and CmdsRNase in Figure 2 represents the product of the gene, namely the zymoprotein. 5. Figure legends (8-10) should be improved to give readers better explanation of results (treatments, insect name etc.) Response: We revised the legends of Figures 8-10. Please kindly see Lines 1146-1151, 1152-1155, and 1157-1161 in the revised manuscript. 6. Conclusion should be added. Response: Conclusion has been added and revised. Please kindly see the change in Lines 2318-2323 in the revised manuscript. 7. line 379 E=what is the concentration of PTU and PMSF? Response: 1 mM phenylthiourea (PTU) and 1mM phenylmethylsulfonyl fluoride (PMSF) were used in the experiment. We had corrected this mistake in Line 1948 in the revised manuscript.

Reviewer 2 Report

In this manuscript the authors identify and describe a nuclease that can degrade dsRNA, thus inhibiting RNAi in the rice leaf folder.

This manuscript is well written the the experiments cover all of the necessary steps, however, I found a couple areas where clarification is needed.

Please clarify the RNAi injection part of the methods. How much dsRNA was injected into each insect? What life stage? What kind of needle? Where on the insect?

In the experiments where you look at affects at pupal and adult life stages, when were these individuals injected? As larvae and survived to adulthood?

Author Response

In this manuscript the authors identify and describe a nuclease that can degrade dsRNA, thus inhibiting RNAi in the rice leaf folder. 

This manuscript is well written the experiments cover all of the necessary steps, however, I found a couple areas where clarification is needed.

Please clarify the RNAi injection part of the methods. How much dsRNA was injected into each insect? What life stage? What kind of needle? Where on the insect?

Response: We appreciate your valuable comment. We have revised the content, that is “Twenty healthy third-instar larvae were selected for each RNAi experiment in each group. One point five micrograms of dsCmCHS or the mixture of dsCmdsRNase and dsCmCHS were injected into the eighth abdominal segment of each larva using an Nanoliter 2020 Injector (World Precision Instruments, Sarasota, FL, USA), and then the larvae were moved onto fresh rice leaves inside glass tubes and cultured in an artificial climate box”. Please kindly see the change in Lines 2147-2151 in the revised manuscript.

In the experiments where you look at affects at pupal and adult life stages, when were these individuals injected? As larvae and survived to adulthood?

Response: The third-instar larvae were injected with dsRNA for RNAi. The surviving larvae could develop into pupae and adults, but some of the pupae and adults exhibited deformed morphology.